# Damage Detection for Rotating Blades Using Digital Image Correlation with an AC-SURF Matching Algorithm

**DOI:** 10.3390/s22218110

**Published:** 2022-10-23

**Authors:** Jiawei Gu, Gang Liu, Mengzhu Li

**Affiliations:** 1School of Civil Engineering, Chongqing University, Chongqing 400045, China; 2The Key Laboratory of New Technology for Construction of Cities in Mountain Area of the Ministry of Education, Chongqing University, Chongqing 400045, China

**Keywords:** rotating blades, damage detection, digital image correlation, angle compensation, speeded-up robust features, dynamic strain

## Abstract

The motion information of blades is a key reflection of the operation state of an entire wind turbine unit. However, the special structure and operation characteristics of rotating blades have become critical obstacles for existing contact vibration monitoring technologies. Digital image correlation performs powerfully in non-contact, full-field measurements, and has increasingly become a popular method for solving the problem of rotating blade monitoring. Aiming at the problem of large-scale rotation matching for blades, this paper proposes a modified speeded-up robust features (SURF)-enhanced digital image correlation algorithm to extract the full-field deformation of blades. Combining an angle compensation (AC) strategy, the AC-SURF algorithm is developed to estimate the rotation angle. Then, an iterative process is presented to calculate the accurate rotation displacement. Subsequently, with reference to the initial state of rotation, the relative strain distribution caused by flaws is determined. Finally, the sensitivity of the strain is validated by comparing the three damage indicators including unbalanced rotational displacement, frequency change, and surface strain field. The performance of the proposed algorithm is verified by laboratory tests of blade damage detection and wind turbine model deformation monitoring. The study demonstrated that the proposed method provides an effective and robust solution for the operation status monitoring and damage detection of wind turbine blades. Furthermore, the strain-based damage detection algorithm is more advantageous in identifying cracks on rotating blades than one based on fluctuated displacement or frequency change.

## 1. Introduction

Blades are critical components in structures that directly determine power generation efficiency and the lifespan of wind turbines. Unfortunately, since the blades are exposed to high-altitude harsh environments and tend to suffer from sudden climatic disasters such as lightning strikes and strong winds [1], they are the part most susceptible to damage in wind turbines [2]. More importantly, blades are bendable and flexible owing to its cantilever characteristic [3], so it needs to bear inertial loads caused by rotation, intermittent wind loads, and large self-weight during operation [4]. The preventive monitoring of blades during operation contributes to reduced downtime and maximized wind farm yields rather than shutdown visual inspection. Damage detection in advance contributes to predicate pertinent maintenance on fault extension trends and avoids catastrophic accidents [5]. It is of great necessity to guarantee the economic benefits as well as prolong the service cycle of wind turbines. Therefore, regular health monitoring and damage assessment of rotating blades have attracted great attention in the engineering community.

Currently, several sensing technologies have been applied to the online monitoring of operating wind turbines [6,7]. Chen et al. [8] collected monitoring signals of acoustic emission systems from commercial wind fields, which were used for health condition assessment of the wind turbine blade (WTB). Based on piezoelectric property, Kang et al. [9] developed a composite blade using piezoelectric materials for real-time collision monitoring and localization. Tcherniak et al. [10] implemented a vibration-based approach by mounting accelerometers on WTBs. During a 3.5-month monitoring campaign, it was observed that a trailing-edge opening gradually increased from 15 cm to 45 cm. Sierra-Pérez et al. [11] compared fiber brag gratings with conventional strain gauges in the strain hotspot monitoring of WTBs and manifested their merits on the detection of premature defects. Since the aforementioned approaches are based on contact sensors, three common flaws arise when applied to rotating blades: (i) A sensor attached on the external surface of the blade is prone to wire looseness and debonding and those pre-embedded into blade during the manufacturing process may induce initial defects [12]. (ii) The replacement of a broken blade-embedded sensor is complicated and requires the disassembling of the blade [13]. (iii) Contact-based discrete sensors only detect local damage, which leads to inadequate monitoring [14]. On the contrary, emerging non-destructive testing (NDT) such as infrared thermography [15], laser Doppler vibrometer (LDV) [16], and photogrammetry [17] provide a cost-effective alternative. Contrasting with point-wise sensors, NDT is characterized by full-field measurement without the installation of sensors and easy implementation [18]. For instance, LDV is a suitable method for blade monitoring since it allows measurements of its surface vibration [19]. Chen et al. [20] proposed an experimental modal testing the technology of WTBs using 3D LDV. The two surfaces of the blade are scanned in a non-contact fashion to characterize the mode shapes of the blade. Lyu et al. [21] developed an image-based tracking scanning LDV system. The edge of a rotating fan blade is located using edge detection methods for navigating the scan path of laser spots. Additionally, it estimates its modal parameters from the measured data of the rotating fan blade. NDT seemingly provides a possible solution for the monitoring of the WTB. However, limited to the large size of blades, the existing NDT is still in the stagnant period of stationary inspection of the local region [22], such as production evaluation and shutdown maintenance. Thus, exploring a comprehensive methodology that enables remote, full-field, and noncontact monitoring of rotating blades is desired.

In recent years, image-based measurement techniques have been extensively investigated for WTB inspection and condition assessment [23,24]. Ozbek et al. [25] conducted on-site tests on an 80 m diameter WTB using four cameras from a measurement distance of 220 m. The vibration responses of optical targets marked on operating WTBs were monitored and then compared with those of strain gauges and laser interferometry. The results demonstrated the feasibility of photogrammetry substituting the other two types of sensing instrumentation. Baqersad et al. [26] adopted point-tracking to identify motion at discrete locations while blades were rotating. On this basis, combined with the finite element model, the abnormal loading situations were detected by predicting the full-field dynamic strain of blades. Considering the difficulty in modeling the geometry and composite materials of WTBs, Bharadwaj et al. [27] extracted strain-mode shapes of blades using digital image correlation (DIC). Poozesh et al. [17] evaluated the performance of a multi-camera DIC system on inspecting large areas over a 50 m blade. Through full-field strain information during static and fatigue tests, it was proved that DIC has the potential to monitor utility-scale WTBs. Wu et al. [28] applied 3D-DIC on a 5 kW WTB to verify the applicability of monitoring the health of a rotating blade by its dynamic strain data. Furthermore, damage detection based on strain characteristic of WTBs was researched with DIC in the laboratory [29,30].

However, strain-based damage detection with DIC is only available for indoor quasi-static tests. There are few reports describing the performance of applying DIC on to an operating wind turbine. It is necessary to validate the practicability and detectability of the DIC, as well as the sensitivity of strain as a damage detection factor. To this end, based on the analysis of the operating structural characteristics of the rotating blades, this paper proposes a modified speeded-up robust features (SURF)-enhanced DIC algorithm. Combining an angle compensation (AC) strategy, the AC-SURF algorithm is developed to estimate the rotation angle. Then, with reference to the initial state of rotation, the relative strain distribution caused by its flaws is determined and used to recognize any damage.

This paper is organized as follows. Section 2 describes the modified SURF-enhanced DIC algorithm. Section 3 details the model experiment setup and results. Section 4 presents the discussion. Section 5 concludes.

## 2. The Modified SURF-Enhanced DIC Algorithm

### 2.1. The AC-SURF Algorithm

The conventional DIC is a subset-based image matching method that determines the target location by comparing the correlation degree between the reference and deformed subsets [31]. It is notable that it has the limited convergence range of a rotation angle; this is a decorrelation problem that appears when the deformed image is subjected to large-scale rotation [32]. In addition, large rigid body translation is predominant in the target subset under the presence of periodic rotation motion. While the deformation of a measured object is relatively small, it would be neglected by mistake. To present the strain distribution of large inter-frame deformation, accurate removal of the rigid rotation is a primary task [33]. Hence, the rotational angle of subsets is needed to be determined in advance.

A modified SURF technique, abbreviated as AC-SURF algorithm, is proposed to estimate the rotary angle and efficiently eliminate the impact of arbitrary rotation changes, as shown in Figure 1.

Firstly, this method uses SURF to match feature points between the reference and deformed images. In general, the correctness of the initial guess estimation determines the convergence speed and precision of the subpixel registration [34]. Thus, the feature point pairs are demanded to a high similarity degree. Therefore, an evaluation criterion based on a customized ring template is presented to quantify the similarity of matched points. High-correlated point pairs are screened to calculate rotary angle *θ*_1_ of two images. Next, an angle compensation approach turns SURF into iterative form by adversely rotating the deformed image according to the calculated rotary angle. Additionally, the temporary rotated image is matched with the reference image. Rotary angle *θ*_2_ is gained and then used to invert the deformed image using angle compensation once again. Repeat the above steps and terminate iteration once the calculated rotary angle satisfies the convergence condition. Finally, the exact rotary angle of reference and deformed images are obtained by the accumulation of the calculated rotary angle (*θ*_1_, *θ*_2_,…, *θ_t_*, the subscript *t* denotes the iteration number). The exact angle herein is extracted to initialize the deformed image as a candidate for subsequent DIC analysis. As Figure 1b illustrated, the ring template consists of concentric rings whose greyscale values are calculated as follows:(1)$$\{\begin{array}{c} P(r_{i})=\frac{1}{m_{i}}\sum_{j=1}^{m_{i}}f_{j}(x,y)\\ Q(r_{i})=\frac{1}{m_{i}}\sum_{j=1}^{m_{i}}g_{j}(x,y) \end{array}$$
where *r_i_* represents *i*-th ring (*i* = 1,2,…, *n*), *n* is the total number of the rings, and *m_i_* is the amount of pixel points located in the *i*-th ring. *f_j_*(*x*, *y*) and *g_j_*(*x*, *y*) denote the grayscale value of the *j*-th pixel point inside the ring. *P*(·) and *Q*(·) mean ring template in the reference and deformed images, respectively. Consequently, the similarity of the ring template in reference and target subsets can be written as
(2)$$S=-\frac{1}{2}\sum_{i=1}^{n}{\left(\frac{P(r_{i})-\bar{P}}{\sqrt{\sum_{i=1}^{n}{\left(P(r_{i})-\bar{P}\right)}^{2}}}-\frac{Q(r_{i})-\bar{Q}}{\sqrt{\sum_{i=1}^{n}{\left(Q(r_{i})-\bar{Q}\right)}^{2}}}\right)}^{2}+1$$
where *S* is the correlation coefficient of two subsets, which falls into [−1, 1]. $\bar{P}$ and $\bar{Q}$ are the mean greyscale intensity of ring template.

### 2.2. The AC-SURF-Enhanced DIC Algorithm

Owing to angle compensation, it enables DIC algorithm to tackle deformation with high precision. The integer–pixel displacement of the target point in the deformed image is searched out pixel-by-pixel, and subpixel registration is performed by using an IC-GN iterative algorithm. In Figure 1d, the off-center position Δ*x*, Δ*y* in the reference image localizes target point. Additionally, then the IC-GN algorithm with the first-order shape function is employed to extract the subpixel displacement. In each iteration, it exerts an incremental parameter Δ*p* on the reference subset and inverts the incremental shape function W(*p*; Δ*p*) to integrate with shape function. Then, the updated function W(*p*; *p*)∙W^−1^(*p*; Δ*p*) is applied to the deformed subset and intensity interpolation is performed to estimate the subpixel locations. The iteration ends until the parameter increment of two subsets satisfy the convergence condition. The detailed iterative procedure refers to the literature [35].

The location of the target point (*x’*, *y’*) in the deformed subset is determined by *p*(*u*, *v*, *u_x_*, *u_y_*, *v_x_*, *v_y_*)^T^ and its final displacement (Δ*u*, Δ*v*) can be obtained by,
(3)$$\left\{ \begin{array}{l} \Delta u=x^{\prime }-x=u+u_{x}\Delta x+u_{y}\Delta y+u_{r}\\ u_{r}=\Delta x\cos\theta +\Delta y\sin\theta \\ \Delta v=y^{\prime }-y=v+v_{x}\Delta x+v_{y}\Delta y+v_{r}\\ v_{r}=-\Delta x\sin\theta +\Delta y\cos\theta \end{array}\right.$$
where *θ* is the exact angle calculated by angle compensation, *u_r_* and *v_r_* represent displacement induced by rigid rotation.

### 2.3. Full-Field Deformation and Strain Measurement

After the precise location of the target point is determined, its deformation parameters are transferred as the initial guess of its neighboring points [36]. As shown in Figure 1d, the calculated target point is regarded as a seed, adjacent four points are then optimized using the IC-GN algorithm. Grid step Δ*L* defines the interval of adjacent points. The one having the maximum correlation coefficient is chosen as a new seed, meanwhile the others are sorted by similarity into a database of calculated points. Three points around the new seed are calculated in the same way. A new seed point will be screened out from database according to correlation degree. Afterward, remaining points are calculated consecutively. In this scenario, full-field deformation is derived until the database is cleared. With reference to the initial state of rotation, the relative strain distribution caused by flaws is determined.

### 2.4. Simulated Tests

To validate the performance of the proposed modified SURF-enhanced DIC algorithm, two simulation tests were carried out using synthetic speckle images. One is to simulate the large deformation of speckle images during the rotation process and test the applicability of the proposed method for deformation measurement in complex situations. Another is to simulate the situation where edge cracks appear and gradually expand on the rotating blade, and then verify the ability of the proposed algorithm to identify abnormal deformation caused by cracks.

The traditional SURF algorithm was introduced to contrast with the proposed algorithm. Computation tasks were conducted by Matlab and a desktop computer (Inter(R) Core (TM) i5-7400 CPU and 8 GB RAM).

Simulation 1: The reference image with a size of 400 × 400 pixels consists of 3600 Gaussian speckles of a 2.4 pixels size. Sinusoidal displacement *v* along *y*-direction is used as a local deformation and added into the reference image to construct the deformed image. The sinusoidal amplitude is set as 1 or 5, representing small and large deformation cases, respectively, as shown in Figure 2a. Then, a series of deformed images are generated by rotating the deformed image clockwise around its center. Additionally, the rotational angle successively increases from 10 to 180 degrees by increments of 10 degrees.

The area inside dot yellow lines in Figure 2b, is selected as a region of interest (ROI). A bicubic interpolation algorithm is employed to determine the subpixel location. A rectangular subset with size of 21 × 21 pixels is used for DIC analysis. The radius of the ring template is set to 9 pixels and the width of each ring is 1 pixel. In the current test, matched points whose correlation coefficient *S* < 0.9995 are eliminated. In the large deformation case, the threshold of *S* is 0.995. If the rotational angle increment is less than 1 × 10^−4^ or the number of iterations is up to 3, the angle compensation ends.

Figure 3 depicts the measurement results of simulated speckle images with two different degrees of deformation. The point with the local coordinate (200, 50) is chosen as the detected target. Each pre-assigned rotational angle is calculated repeatedly 200 times. Both mean biases are measured by two algorithms and the standard deviations symbolized by error bars are plotted in Figure 3a,b.

From the above two deformation cases, it could be seen that biases of the proposed method are distinctly smaller than those of the traditional SURF algorithm. Although large deformation leads to the amplitude of the error increases, the proposed algorithm maintains the error in a small and stable range. In particular, the error bars indicate that results using the SURF algorithm in Figure 3a fluctuate randomly, whereas those in Figure 3b are close to zero. This is because deformation destroys the inherent features of an image. When the deformation is small, many unstable feature points will be matched and result in high randomness in the estimation of the transformation model of the two images. After undergoing large deformation, the standard deviations reduce as the number of feature points decrease. In comparison, it is worth observing that the proposed algorithm turned out to be robust against the variations of image features. In practice, evaluate the criteria benefits to accurately extract the best-matched points and get a reliable and exact rotational angle.

Randomly select a pixel point in the ROI as an initial seed point and conduct full-field measurement with a grid step Δ*L* of 3 pixels. For conciseness, Figure 3c,d shows the full-field results measured by the proposed algorithm in *y*-direction for the large deformation case corresponding to a 40° rotational angle. In Figure 3d, the strain calculated is a Green–Lagrange strain. The ZNCC coefficients are all greater than 0.99, which proves the high confidence of the proposed algorithm.

To compare the performance of two algorithms in terms of the full-field displacement measurement in the large deformation case, the root mean square (RMS) error is used to quantitatively assess the total bias level.
(4)$$\mathrm{RMS}=\sqrt{\frac{\sum_{i=1}^{N}{\left(d_{i}-{d_{i}}^{\prime }\right)}^{2}}{N}}$$
where *d_i_* and *d_i_’* are the measured displacement and real value at each calculation point. *N* is the count of points in a grid.

There are noticeable biases in the displacement fields measured by the conventional method in Figure 4. The resultant deformation in *x*-direction is composed of continuous elongation and compression, and its change trend shows the opposite with the preset sinusoidal displacement along *y*-direction. Furthermore, the displacement fields estimated by the traditional method have a global translation compared to the zero horizontal line plotted by black dashed box. This is mainly due to the influence of initial parameters in the process of propagation. The errors in Figure 3a,b resulting from the conventional algorithm will be transferred to its neighboring points.

In contrast, the proposed method is capable of achieving a relatively accurate initialization of deformation parameters, thus greatly alleviating the computational cost and reducing the total bias of the displacement field.

Simulation 2: In order to detect the strain concentration on the blade edge caused by cracking during rotation, this test uses weld tensile images (SEM-Sample13) from the Society for Experimental Mechanics (https://sem.org/dicchallenge, accessed on 15 May 2021) as the speckle pattern on the blade surface. These images record the expansion of cracking with the specimen stretching.

The Sample13 images are intercepted by blade images with a size of 710 × 710 pixels, the crack region is overlapped onto one blade. Additionally, the deformed images are then rotated clockwise with an incremental angle of 45° as shown in Figure 5. The full-field deformation of the rotating blade is reconstructed by the proposed algorithm as shown in Figure 6.

Due to the full-field displacement of blade changes with the increase of the rotation angle, the abnormal deformation of blades caused by cracking is covered by the large-scale displacement. The separation of large-scale rotation is an important premise in detecting strain concentration for rotating blades. The strain component of the blade is separated using the angle compensation algorithm as shown in Figure 6b. Strain concentration exists in the blade tip, and it increases with the rotation of the blade. It is concluded that: (a) the damage position is expected to be accurately detected according to the dynamic strain field, and (b) regular monitoring for the rotating blade using the proposed algorithm is effective for the judgement of the damage growth trend.

With the purpose of accurately locating the damage, the strain curve along the length of blade is extracted in Figure 6c. The change of strain at the blade root is stable, but it increases abruptly near the blade tip, where the location of weld cracking is. According to the abnormal peak value of the strain curve, the damage location of the rotating blade is ascertained.

## 3. Experiments and Results

To further demonstrate the applicability of the proposed method, a wind power generator model was built in a laboratory. The manufacture of blades was adapted using injection-molding technology. The characteristics of blade damage caused by cracking are first extracted by a static load test, and then are used to monitor any abnormal deformations during the rotation of the wind turbine blade.

### 3.1. Damage Detection of Cantilevered Blade

The composite blades are constructed using steel sheets as the core material and rubber as the layer for the demand of flexibility. The blade is 390 mm in length and 30 mm in width at the blade tip. The surface of the blade was dotted with random black and white spots as a speckle pattern. Cracks of the wind turbine blade are usually distributed in the leading edge of the blade root. In this experiment, an artificial crack along the width of the blade was set at a distance of 104 mm from the blade root. The cracked region of blade is 80 mm in width and 10 mm in thickness. The crack with 1 mm width, 1.5 mm depth, and 30 mm length was set on the backside of the blade. Subsequently, the blade root was fixed using a jack. A loading device was installed near the blade tip to apply vertical deformation as shown in Figure 7. Admittedly, it would be pertinent with real blades, if using an assumed loading or wind blowing, that the blade could move to better simulate the deformation. However, static tests of cantilevered blades are used to be certificated with strain gauges. For the flexible blade, dynamic loading in the edge-wise direction may cause large vibrations in the flap-wise direction. In this scenario, data derived from strain gauges would lose their accuracy and reliability. A static loading test is able to reduce the flap-wise deformation of a blade.

An 8-bit gray level CMOS camera (Daheng MER503-G-P, Beijing, China) with a resolution of 2448 × 2048 pixels was used to capture images; the lens is placed perpendicular to the blade and the distance measurement is 0.34 m. The calculated scale factor is 0.0714 mm/pixel. LED lighting was used to maintain site illumination stability. The standard deviation of image noise during the test is approximately 2.5 gray values. Five strain gauges were equidistantly distributed with a spacing of 25 mm around the crack. A static strain test device (Donghua DH3818N-2, Beijing, China) with the sampling frequency 2 Hz was used to acquire the strain data of the blades. Table 1 shows the loading condition. Herein, the vertical displacement of the blade tip is controlled during loading, whose aim is to cause deformation of the blade.

The full-field strain distribution in the *x*-direction of the blade is reconstructed in Figure 8. It can be observed that the strain at the preset crack exists as a dramatic concentration and turns more obviously as the loading increases. To further locate cracks from the strain field, the strain curve along the length of the blade and discrete data of the strain gauge are shown in Figure 9.

Through the strain curve along the length of the blade, an abnormal peak value occurs at the same position under different loading conditions. It can be determined that the damage is 104 mm away from the blade root, while the strain measured by the strain gauge is discrete and cannot be used to judge damage. It is notable that the difference of strain data of the two methods at the measuring point 5 is large. This is because the out-of-plane bending of the blade during the loading process causes the captured image to zoom-out, thus leading the strain field calculated by the DIC method to be in compression. It is the blade that is subject to the force along the flap-wise direction that yields the uncertainty of measurement. However, this is consistent with the real situation, because the blade is inevitably deformed in the swing direction under the influence of wind loads when the blade rotates. The strain field at the damage will change abruptly but the overall deformation is only caused if there is no damage, the strain changes uniformly, and the judgment of damage will not be negatively affected.

### 3.2. Deformation Monitoring of Rotating Blade

With the purpose of verifying the deformation monitoring ability of the proposed method in the rotating state of the wind turbine blade, the wind turbine model test is carried out in the structural engineering laboratory of Chongqing University. Since the wind turbine blade and the composite material are slender, the design of the scaled model is difficult to meet with similar criterion in terms of material and scale ratio. Therefore, the wind turbine model blade and its crack size in this experiment are the same as that in Section 3.1. The height of the model hub is 1.1 m, and it is driven by a servo motor. The speckle pattern on the blade surface is fabricated with random black and white dots. An 8-bit grey level and 2448 × 2048 pixels CMOS camera (Daheng MER503-G-P, Beijing, China) was used to capture the images at a frame rate of 20 frames. Additionally, the lens of the camera is placed perpendicular to the center of hub with a measuring distance of 1 m. One of the three blades was randomly selected to preset a crack along the width of blade and the crack is 123 mm from the center of the hub. The view field of the camera was limited in a local region within 240 mm from the center of the hub. Set the motor constant speed 15 r/min, 20 r/min, and 30 r/min as the three experimental cases. Each test condition was shot for 10 s, and the number of captured images was 200 frames. The standard deviation of image noise in this experiment is approximately 3 grayscale values, and the scale factor is 0.1786 mm/pixel. The experimental setup is shown in Figure 10.

Accurately identifying the displacement of the rotating blade is the precondition for the determination of the dynamic deformation field. The image taken by the shutdown state of the WTB is used as a reference image. The pixel point at the preset crack and its two adjacent pixels points along the length of the blade with intervals of 200 pixels (35.7 mm) are used as the three measured points. In order to check the effect of cracks on the blade trajectory, three measured points on a healthy blade with the same length from the center of the hub are selected. Table 2 shows the location of measured points.

Since the vibration of the tower or the motor will be transmitted to the blade, the center of the hub is used as the origin of the calculation coordinate system to extract only the rotational displacement of the blade. Figure 11 shows the displacement time–history curves of the measured points. All estimated displacements in the *x*- and *y*-directions exhibit a similar trend of rotation and only those from the *y*-direction in the first 5 s are discussed.

The correlation coefficients for the correlation matching of rotating blades are greater than 0.99. In the case of low motor speeds 15 r/min and 20 r/min, the movement trajectories of the damaged blade and healthy blade both maintain a stable sinusoidal curve. However, the trajectory of the measured points of the damaged blade fluctuates slightly when the rotation speed reaches 30 r/min. Especially those of the measured points 4 and 5 are more obvious. These two measured points are located at the crack and outside of it, whereas the results of measured point 6 inside the crack is not affected by the rotational speed, which is consistent with the expected results. The centrifugal force of the blade increases with the rotation speed, which will cause the stress concentration in the area with a crack. Besides, the combined effect of self-weight and centrifugal force varies at different rotational orientations of the blade, so the fluctuation of the motion trajectory presents a periodic change. It is because low rotational speed is not enough to cause a large vibration at the crack relative to the rotational displacement. However, when the rotational speed increases, this vibration expands to a visible change in the motion trajectory.

Frequency change is a common sensitive index reflecting structural damage [37,38]. The Fourier transform was performed on the *x*- and *y*-direction dynamic displacements of six measured points in 10 s. The results of the spectrums are shown in Figure 12. However, the frequency index of the two tested blades remained the same, only a fundamental frequency signal caused by rotation exists, without other harmonic components from the crack. Therefore, the preliminary damage of the rotating blade cannot be figured out in this way.

Considering the gravity and clockwise rotation direction of the blade, the horizontal right orientation is chosen as the worst load condition. Additionally, the image of the blade captured in this orientation is used as the deformation image. Using the proposed method, the *x*-direction strain distribution with respect to the reference position is shown in Figure 13. The background images in Figure 13 are reference images of each case.

Similar to the motion trajectory plotted in Figure 11, the strain of the blade is supposed to be time-variant and periodic. The location where the strain concentration of the blade occurred, at the same location at a fixed rotation orientation, could be damaged. For the healthy blade, the reference images of three cases are in different states and the horizontal right orientation of the deformed image cannot be completely consistent. So, the strain distributions of healthy blades are slightly different. However, the strain field was continuous, and no strain concentration or local similarity appeared, which is in accordance with the actual situation. In contrast, the alike characteristics were present in the cracked region of the damaged blade in three cases. There is an obvious strain segmentation along the length of the crack and this phenomenon becomes more striking with increasing rotational speed.

Moreover, the strain distributions along the length of the damaged blades were extracted for further damage analysis, as shown in Figure 14. The strain abnormal peak occurs at the same position, approximately 123 mm from the center of the hub. The position error does not exceed 1 mm. Nevertheless, this level of error already meets the measurement requirements, because the blade will be deformed by centrifugal force during rotation. In this case, the real position of the preset crack will also slightly change.

## 4. Discussion

The performance of the proposed algorithm is verified by two experiments on a wind turbine model. The blade itself is flexible and has a large deformation during the operation. The deformation of a healthy blade changes uniformly, but it leads to deformation concentration due to changes in the cross-section where the crack exist. It is justifiable to detect cracks using the surface strain field. In the blade damage detection test, the surface strain of the blade with a crack is reconstructed in Figure 8. The strain in a non-destructive area changes uniformly, with an obvious strain concentration at the crack position, and it becomes more concentrated as the magnitude of the external load increases. Therefore, regular monitoring of the damaged blade is necessary to judge the degree of damage and its trend of expansion as shown in Figure 9.

In the rotating blade monitoring test, the deformation of the blade becomes more complex under unpredictable loading conditions. Three damage indicators including unbalanced rotational displacement in Figure 11, frequency change in Figure 12, and surface strain field in Figure 13 were compared. The crack divides the blade into two parts, one is the region from crack to root and the other is from the crack to the tip. The vibration caused by the crack provokes the second part’s edge-wise motions, resulting in the blade motion trajectory fluctuating. The further away from the crack, the greater the amplitude of fluctuation. For example, when the speed reaches 30 r/min, the fluctuation of measured point 4 is more obvious than that of measured point 5. However, this slight fluctuation is easily covered by other factors, such as wind load, abnormal vibration of the motor, etc. It is difficult to find out what leads to unbalanced rotation trajectory. Using the frequency change between two blades is a quick tool to distinguish the damaged blade. In Figure 12, the measured displacements were analyzed in the frequency domain, but no difference was found in the spectral curves of healthy and damaged blades. It is possible that the early weak cracks have little effect on the dynamic characteristics of the rotating blade.

The strain fields of blades are reconstructed by the proposed algorithm in Figure 13, and characteristics similar to Figure 8 are located in the same area when the damaged blade is in horizontal orientation. This apparently differs from the strain field of a healthy blade. An increase in rotational speed could change the strain of healthy blade, but its distribution remains continuous. Even though some areas may have concentrated strain due to local load, there is no abnormal peak in the strain curve. Once the blade with a crack is stressed, the strain at the crack is distributed along the shape of crack and is more concentrated than other parts without a crack. Finally, the location of the crack can be ascertained by extracting the strain curve of the damaged area as shown in Figure 14. The deformation of the healthy blade and the strain in non-destructive areas change uniformly. In contrast, an obvious strain concentration at the crack position exists in the blade with a crack. Therefore, it is feasible to detect a crack using the strain field on the blade surface and obtain the location of the crack by extracting the strain curve.

Combining a high correlation coefficient, accurate sinusoidal displacement, sensitive distribution, and curve of strain, the proposed modified SURF-enhanced DIC algorithm was demonstrated to be reliable for the monitoring and damage detection of rotating blades. However, there are some obstructions in the study worth trying to solve. Due to the limited resolution of the camera equipment, only the blade root of the wind turbine model was captured and merely the local region circled by red line was calculated as is shown in Figure 13. Shooting the entire blade is beneficial for comprehensive safety evaluation of blades [39], such as monitoring the flap-wise displacement of the blade tip. If high-resolution cameras are used, a more continuous signal will be obtained and damage can be visualized in more detail [40]. Real WTBs have a huge structure and complex materials, so it is hard to design an accurate scale model in a laboratory. A standard wind turbine unit scale model has the ability to be applied to research on the mechanic properties of actual blades in the case of wind tunnel tests. It is of great importance for the extraction of sensitive damaged indicators under different simulated wind blowing and dynamic loadings. If advanced signal processing methods (such as band-pass filtering and time-frequency transformation) are performed on the displacement or strain, it enables the extraction of more sensitive damage indicators [41]. At present, measurements of WTBs on the ground are not suitable for wind farms; a promising alternative is to use an unmanned aerial vehicle photogrammetric technique [42], or install the camera on the nacelle of a wind turbine to monitor any other WTBs behind itself.

## 5. Conclusions

Aiming at the operating wind turbine, this paper proposes a modified SURF-enhanced DIC algorithm to extract the displacement response and full-field deformation of blades. Then, with reference to the initial state of rotation, the relative strain distribution caused by flaws is determined. Finally, the sensitivity of the strain is validated by comparing the three damage indicators including unbalanced rotational displacement, frequency change, and surface strain field. The performance of the proposed algorithm is verified by laboratory tests of blade damage detection and wind turbine model deformation monitoring. The results showed that the strain field has the potential to accurately determine crack positions, while the other two are inclined to be submerged by environmental factors. It demonstrated that the proposed method provides an effective and robust solution for the operation status monitoring and damage detection of wind turbine blades.

Currently, the method was only applied to a wind turbine model in comparison to an in-site operating blade. The measurement accuracy may be affected by the image acquisition condition on account of the large size of the blade. In the case of the application of this method, a suitable measurement facility is the breakthrough of the blade monitoring for the wind turbine in service. It includes aerial photography and the super-resolution method using a camera array for the limited photogrammetric equipment.

## Figures and Tables

**Figure 1 sensors-22-08110-f001:**
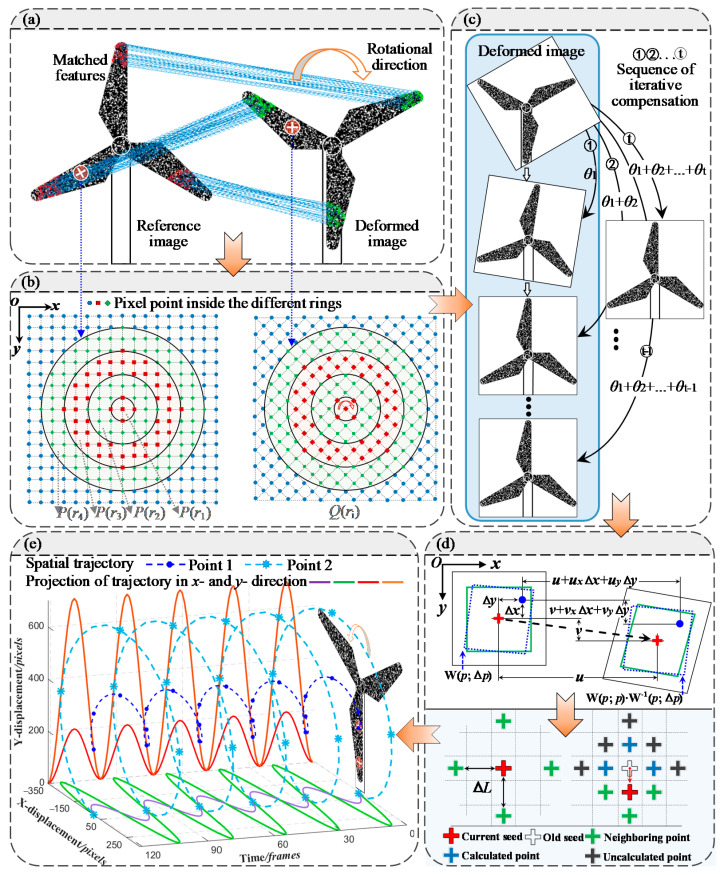
Framework of the proposed AC-SURF algorithm: (**a**) Feature matching using SURF; (**b**) Evaluation criteria using ring template; (**c**) Angle compensation; (**d**) Deformation estimation using DIC; (**e**) Multipoint tracking.

**Figure 2 sensors-22-08110-f002:**
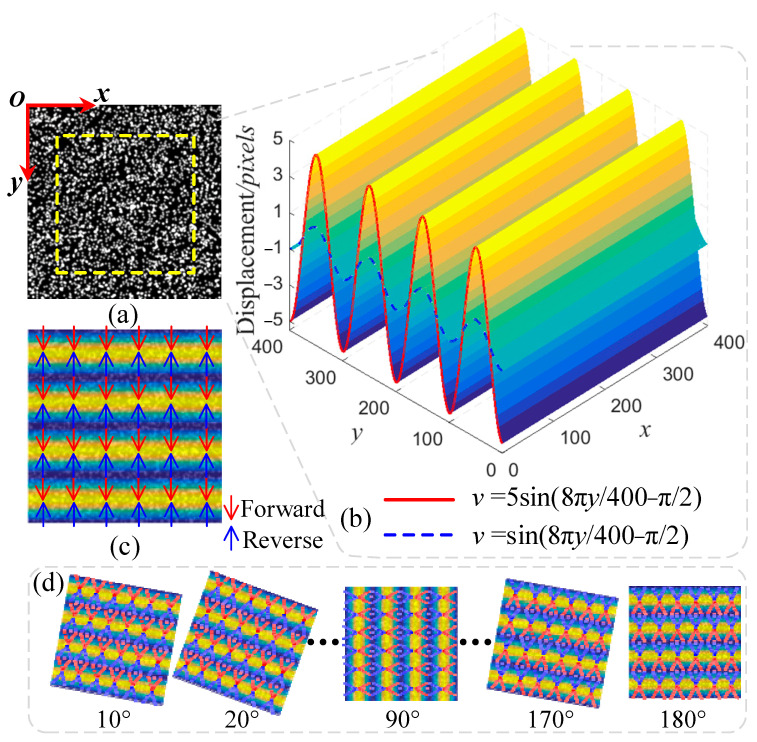
Theoretical deformation field along the *y*-direction mapped onto numerical images: (**a**) Reference image; (**b**) Simulated sinusoidal displacement field in y-direction; (**c**) Deformation image; (**d**) A series of deformed images with increasing rotation angles.

**Figure 3 sensors-22-08110-f003:**
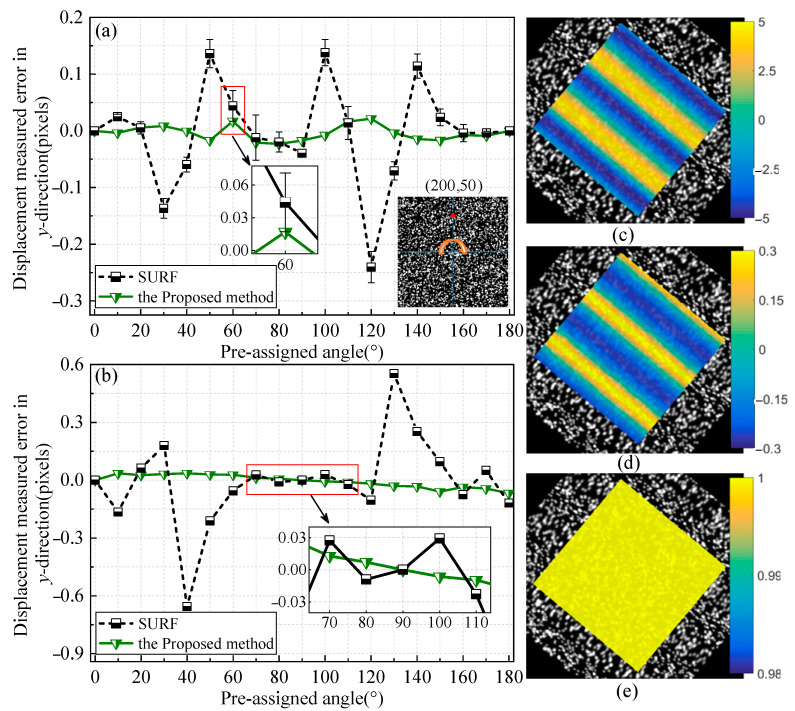
Single-point displacement measured results with mean biases and standard deviations: (**a**) The small deformation case; (**b**) The large deformation case; deformation case. Full-field measurement results at 40° rotational angle in the large deformation case: (**c**) Displacement field in y-direction (pixels); (**d**) Strain field in y-direction (pixels); (**e**) ZNCC value.

**Figure 4 sensors-22-08110-f004:**
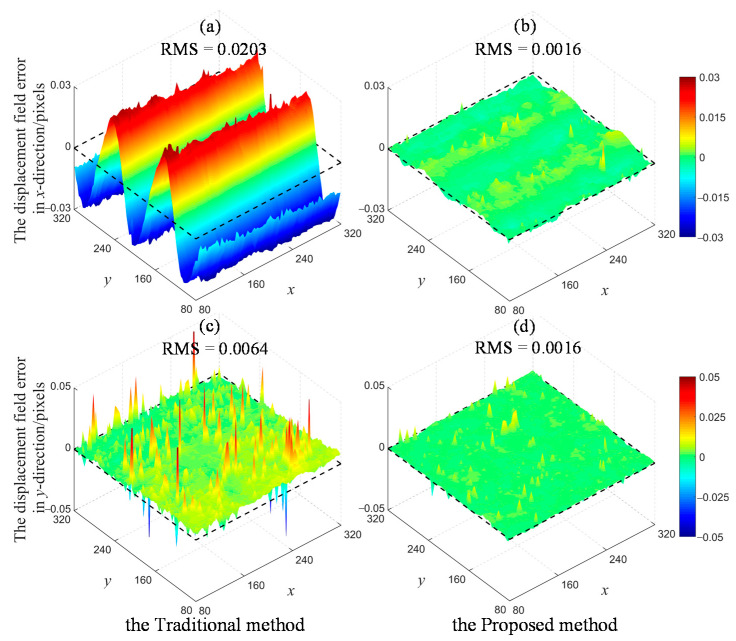
Full-field displacement errors of experiment 1 at 40° rotational angle for the large deformation case: (**a**) The *x*-directional errors obtained by the traditional method; (**b**) The *x*-directional errors obtained by the proposed algorithm; (**c**) The *y*-directional errors obtained by the traditional method; (**d**) The *y*-directional errors obtained by the proposed algorithm.

**Figure 5 sensors-22-08110-f005:**
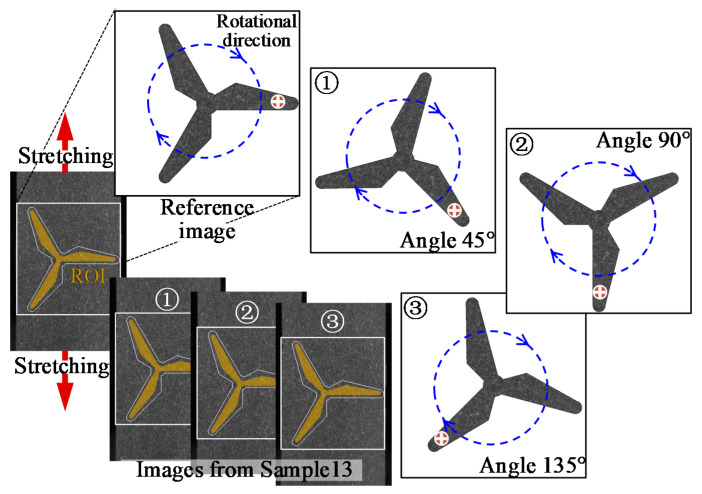
Schematic diagram of simulated rotating blade.

**Figure 6 sensors-22-08110-f006:**
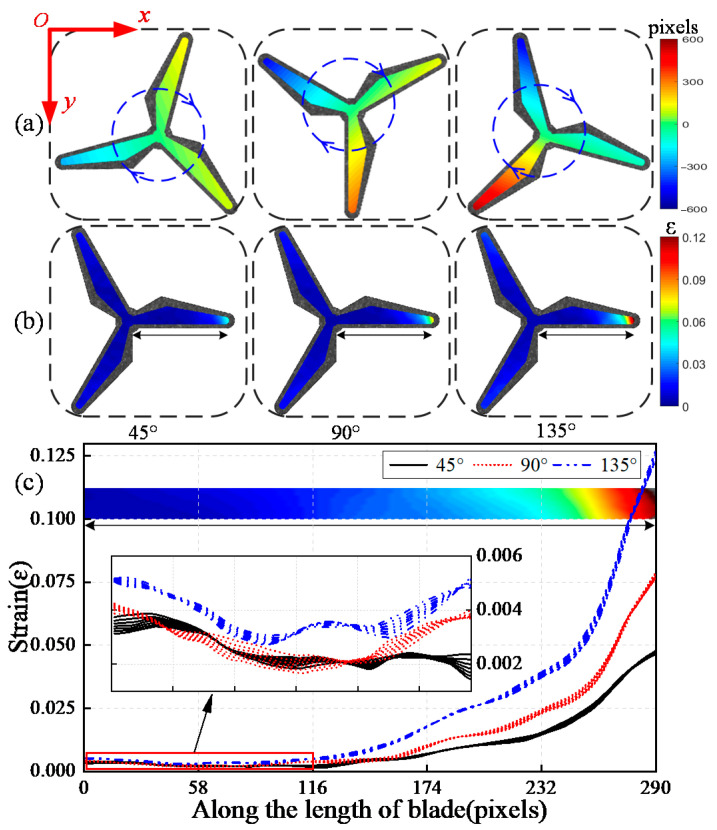
Distribution of full-field deformation in *y*-direction: (**a**) Displacement; (**b**) Strain; (**c**) Curve of strain.

**Figure 7 sensors-22-08110-f007:**
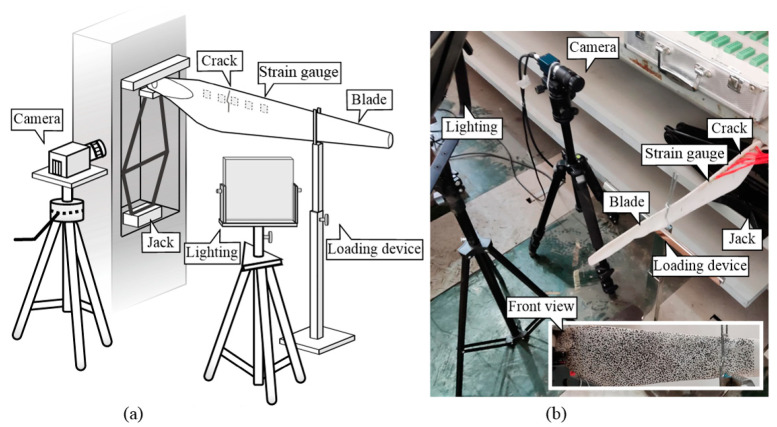
Blade damage detection setup: (**a**) Experimental setup; (**b**) Test site.

**Figure 8 sensors-22-08110-f008:**
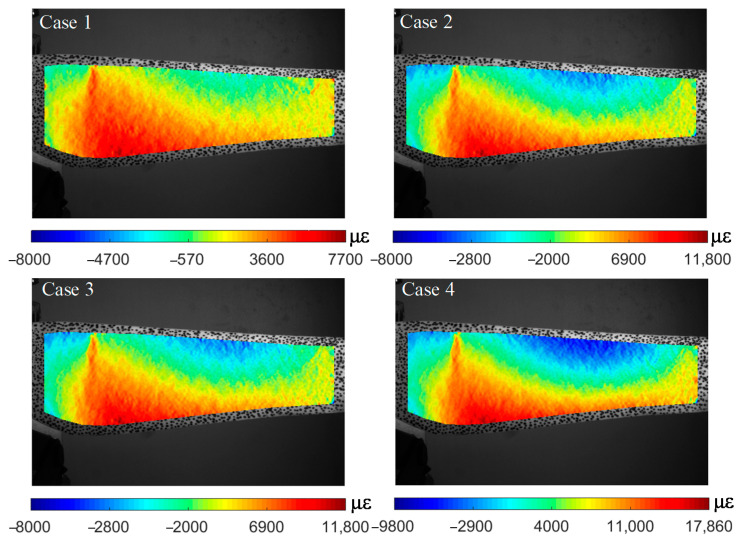
Full-field strain distribution of the blade at different cases.

**Figure 9 sensors-22-08110-f009:**
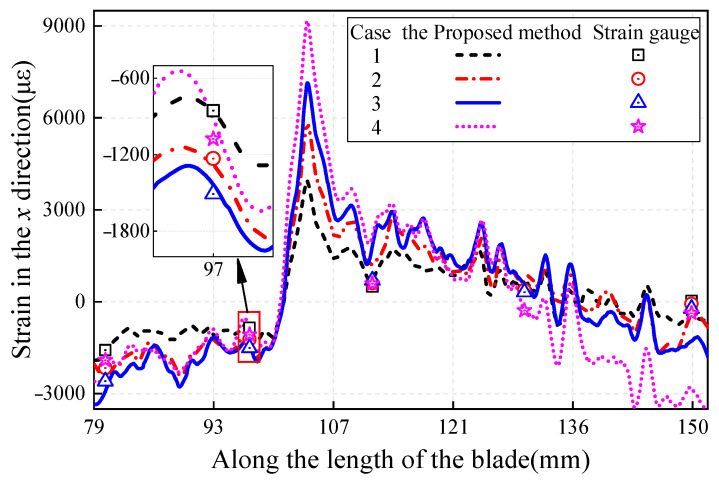
Strain curve along the length of the blade.

**Figure 10 sensors-22-08110-f010:**
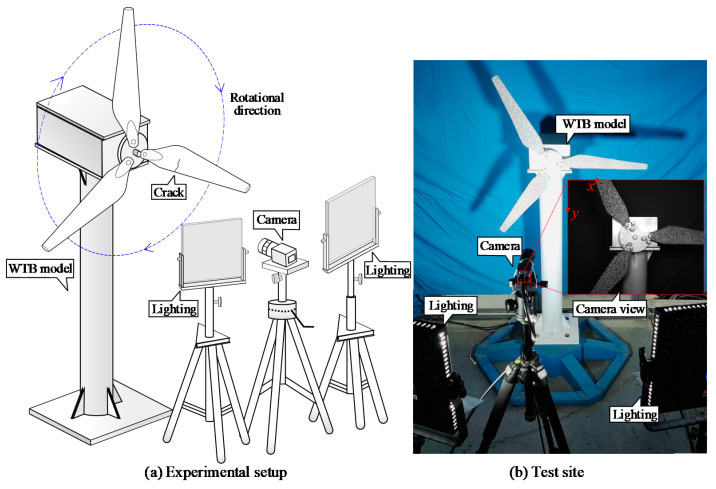
Deformation monitoring of rotating blade setup: (**a**) Experimental setup; (**b**) Test site.

**Figure 11 sensors-22-08110-f011:**
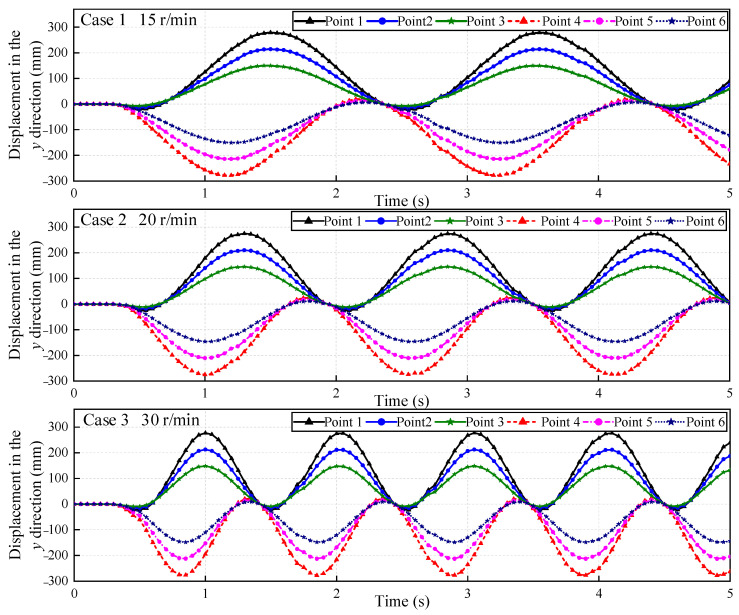
Dynamic displacement of measured points.

**Figure 12 sensors-22-08110-f012:**
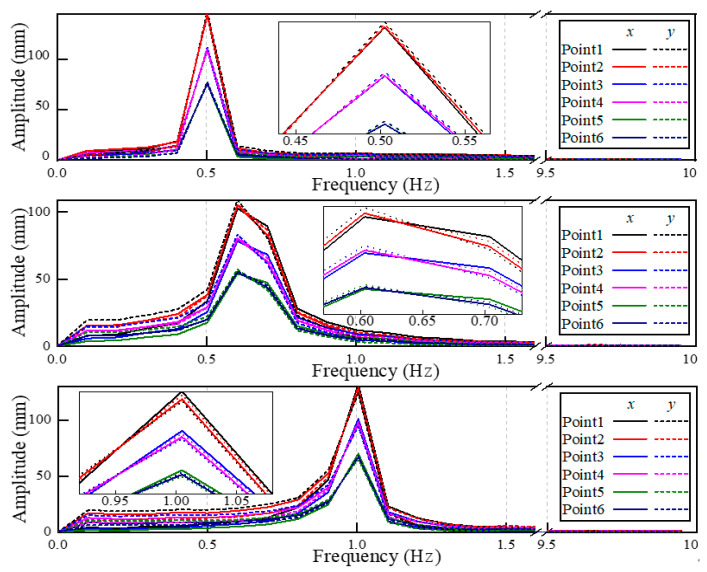
The frequency spectrum of dynamic displacement.

**Figure 13 sensors-22-08110-f013:**
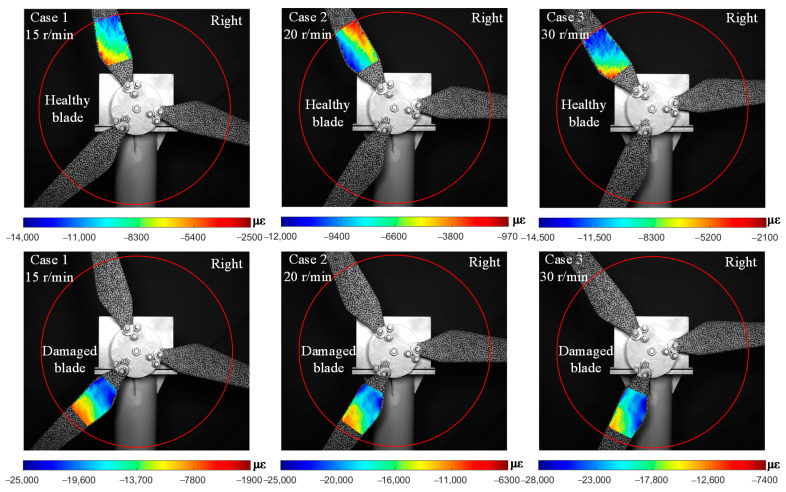
Strain distribution in *x*-direction of the measured blade when rotated to the horizontal right orientation.

**Figure 14 sensors-22-08110-f014:**
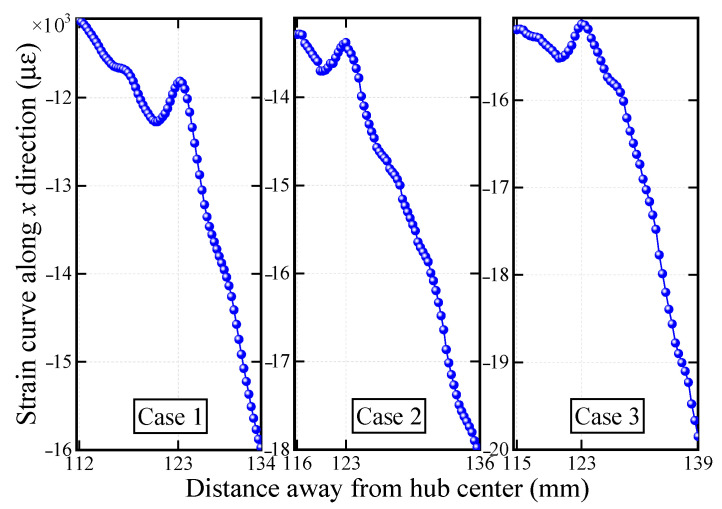
Strain curve of the damaged region.

**Table 1 sensors-22-08110-t001:** Loading condition of experiment 1.

Loading Case	Vertical Displacement (mm)
1	5
2	10
3	15
4	20

**Table 2 sensors-22-08110-t002:** Location of measured points of experiment 2.

Case	Healthy Blade			Damaged Blade		
	1	2	3	4	5	6
1	(797, 305)	(917, 405)	(1037, 625)	(797, 1743)	(917, 1583)	(1037, 1423)
2	(753, 239)	(873, 489)	(993, 649)	(753, 1719)	(873, 1559)	(993, 1399)
3	(775, 315)	(895, 475)	(1015, 635)	(775, 1733)	(895, 1573)	(1015, 1413)

## Data Availability

Data are available on request to the authors.

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
