# Peer review of "Damage Detection for Rotating Blades Using Digital Image Correlation with an AC-SURF Matching Algorithm"

_sensors, 2022, doi:10.3390/s22218110_

Round 1

Reviewer 1 Report

Dear reviewers, I have read this text, here are my comments on the text:

Chapter 3 can be titled results instead of experiments.

Chapter 4 needs to be titled discussion instead of results. And if one were to stick to the original name of the chapter, its content is a misunderstanding. This is because it is very short and not written in detail, and it is usually such a chapter that is the most important part of the text documenting the authors' contribution.

In the case of Chapter 3. however, I have big reservations about this model because the blade is loaded statically, and this time it would be better if the load was assumed so that the blade could move which would better simulate the deformation forces. Another factor is also the testing of bending and torsional stresses which is applicable if we want to simulate wind blowing. These tests should be applied with a model that takes into account dynamic loading. Such testing should take place in a wind tunnel (refers to Section 3.2). Perhaps in this case it is worth including this in the title and in the future expand the experiments by also indicating this in the conclusions.

The discussion (Chapter 4) is poorly written, it should refer to the text in Chapter 3 and discuss the various models used in this work including the interpretation of the evaluation of the results. This is not in this chapter, it is laconic and short and this time he should be closely related to the authors' interpretation of the results.

Part of the text from Chapter 5 (conclusions) can be moved to Chapter 4 (discussion), but in Chapter 5 you also need to remember to summarize the research in the conclusions and show its limitations, further direction of its development.

A small amount of literature: a mere 26 productions.

In general, this text has a lot of potential and I think it is worth publishing, but I ask the authors to improve it.

Reviewer 2 Report

Dear authors,

In the beginning, I would like to thank you for updating the paper. However, there are some issues that I want to point out in order to improve the paper. The quality of the information is especially important for a case study type of paper.

A)    General remarks

1.      Please check if the affiliation style is corresponding to the journal template. Some information may be missing (e.g address).

2.      In the case of the title and the abstract, those elements must be understandable to none familiar to the field reader. It is advised to use no abbreviations if possible. The reviewer suggests at least not using the DIC abbreviation in the title so the title is clear. In the abstract, the authors are using abbreviations, which should be avoided in the abstract and those abbreviations should be introduced for the first time in the main text.

3. Authors are using conditional forms extensively which should be avoided in scientific text. Phrases like “can”, and “could”, should be avoided if possible. Please check if there are occurrences of unjustified use of conditional form.

4. Digital Image Correlation is the name of the technique. Thus it should be written in capital letters. In the text, it is often written one way or another. Please unify to the correct form. E.g lines 11, 76 and keywords section

5. The introduction part is written clearly. However, the reviewer would ask to introduce some other optical techniques that may be possible to use.  The most important one, which is not mentioned, is 3D Laser Doppler Vibrometry. The primary usage is vibration measurements of structures and systems. But with a special direct option or indirect software feature, it is also possible to measure strain-stress. E.g Polytec vibrometers have those options described here https://www.polytec.com/eu/vibrometry/products/software/stress-and-strain-measurement-package. Here is an example of lightweight truss measurements Guinchard, M. et.al Non-invasive measurements of ultra-lightweight composite materials using laser Doppler Vibrometry system, Proceedings of the 26th International Congress on Sound and Vibration, ICSV 2019. Another application is for quality control which can find applications also for blade structure e.g. https://doi.org/10.1007/s40799-022-00604-2 and specific usage for rotating structures https://doi.org/10.1016/j.jsv.2022.116797 . The system is also widely used for crack detection or delamination problems e.g https://doi.org/10.1016/j.optlaseng.2016.10.022 . On comment in case of comparison of the technique. In the case of DIC, although, this is a nondestructive method, still it is necessary to cover the structure with the spackle pattern (paint). In the case of 3D LDV this is a fully optical method with no additional input (usually) on the structure needed. Consider adding those references. https://doi.org/10.1016/j.optlaseng.2016.10.022 . On comment in case of comparison of the technique. In the case of DIC, although, this is a nondestructive method, still it is necessary to cover the structure with the spackle pattern (paint). In the case of 3D LDV, this is a fully optical method with no additional input (usually) on the structure needed.

6. Methods and experimental design are clear and no additional changes are required.

7. The experimental part of the paper is very interesting and no significant changes are required. The reviewer congratulates the author on very clear graphs and figures.

8. Conclusions are clear. One comment maybe for the future. As pointed out before it would be maybe interesting to cross-check the usage of the algorithm not only with the DIC data but also using 3D Laser Doppler Vibrometry. This technique allows for measurements even up to 100m away from the structure which allows for measurement of whole buildings or in this case blades of real wind turbines. This may be an interesting next step.

B)    Item remarks

It is strongly suggested to improve the paper in the case of graphical content. On most figures the  text input is difficult to follow  or even to read, e.g:

·         Fig.1 most of the text is not visible, especially in figs b and d. Also, this is the author's figure or was it taken from another source? No reference is present for any of the sub-images.

·         Fig.2 suggest enlarging

·         Fig.4 suggest enlarging

·         Fig. 6 suggest putting image c in the same column as a and b and enlarging the figure.  

·         Fig. 14 suggest enlarging

C)    Conclusions

Due to some improvements, the reviewer is marking the paper for minor reviews. Hope the authors will use some suggestions to improve this otherwise very interesting and good quality paper.  

Round 2

Reviewer 1 Report

Dear Editor, I read the text after the correction and the authors' responses. I am satisfied with it. I change my opinion.

I would only ask you to add more citations. 

Author Response

Dear  Reviewer:

     We would like to thank you for the thorough review and your positive comments in this round.

      We have added 7 more citations and the number of all references is 42 in the revised manuscript. The added references are listed as following (the number outside the bracket is the serious number in the revised manuscript),

[1]7. Natili F, Castellani F, Astolfi D, et al. Video-Tachometer Methodology for Wind Turbine Rotor Speed Measurement. Sensors, 2020, 20(24): 7314.

[2]13. Zhang C, Yang T, Yang J. Image Recognition of Wind Turbine Blade Defects Using Attention-Based MobileNetv1-YOLOv4 and Transfer Learning. Sensors, 2022, 22(16): 6009.

[3]14. Lich J, Wollmann T, Filippatos A, et al. Spatially Resolved Experimental Modal Analysis on High-Speed Composite Rotors Using a Non-Contact, Non-Rotating Sensor. Sensors, 2021, 21(14): 4705.

[4]24. Liu G, Li MZ, Mao Z, et al. Structural motion estimation via Hilbert transform enhanced phase-based video processing. Mechanical Systems and Signal Processing, 2022, 166: 108418.

[5]37. Liu G, Gao K, Yang Q, et al. Improvement to the discretized initial condition of the generalized density evolution equation. Reliability Engineering & System Safety, 2021, 216: 107999.

[6]38. Gillich G-R, Maia NMM, Wahab MA, et al. Damage Detection on a Beam with Multiple Cracks: A Simplified Method Based on Relative Frequency Shifts. Sensors, 2021, 21(15): 5215.

[7]39. Luo J, Liu G, Huang Z, et al. Mode shape identification based on Gabor transform and singular value decomposition under uncorrelated colored noise excitation. Mechanical Systems and Signal Processing, 2019, 128: 446-62.

     The modified parts have been marked in blue in the revised manuscript for your references.

Yours sincerely,

Gang Liu